DATA RELEASE

# Sand fly (Diptera: Psychodidae: Phlebotominae) records in Acre, Brazil: a dataset

Rodrigo Espindola Godoy[1,*], Andrey José de Andrade[2],
Paloma Helena Fernandes Shimabukuro[3] and
Andreia Fernandes Brilhante[4]

1 Independent Researcher, Brazil
2 Universidade Federal do Paraná, ACF Centro Politécnico, Jardim das Américas, 81531980 - Curitiba, PR 19031, Brazil
3 Fundação Oswaldo Cruz – Centro de Pesquisas René Rachou, FIOCRUZ, Avenida Augusto de Lima - 1715, Barro Preto, Belo Horizonte, MG 30190002, Brazil
4 Universidade Federal do Acre, Departamento de Ciências da Saúde e Educação Física. Universidade Federal do Acre, Distrito Industrial, Rio Branco, AC 69920900, Brazil

## ABSTRACT

Amazonian phlebotomine sand fly fauna is one of the most diverse in the world. The Amazon region is also the most prevalent for leishmaniasis in Brazil and South America. The state of Acre, in this region, also stands out in terms of the diversity of sand fly fauna, as well as the occurrence of American tegumentary leishmaniasis. In this context, the present dataset comprises a bibliographic review of sand fly species recorded in Acre state, Brazil. A total of 1,094 observations from material citations and two of preserved specimens are presented using 33 variables according to Darwin Core terms. The bibliographic review was performed in PubMed, Google Scholar, SciELO, Biblioteca Virtual em Saúde, and references cited in related scientific articles. Thus, this report will be valuable for further studies on sand flies in Acre and other Amazon states.

**Subjects** Ecology, Biodiversity, Taxonomy

**Submitted:** 28 February 2022

\* Corresponding author. E-mail: rodrigoeg@alumni.usp.br

Preprint submitted at https://doi.org/10.1590/SciELOPreprints.3792

Included in the series: *Vectors of human disease* (https://doi.org/10.46471/GIGABYTE_SERIES_0002)

## DATA DESCRIPTION

Here, we present a dataset comprising a bibliographic review of sand fly species recorded in Acre state, Brazil. A total of 22 articles/books, published between 1964 and 2022, were used to obtain the data. As a result, 1,096 observations (including two from preserved specimens) were recorded for the state of Acre. Records were obtained from eight municipalities, namely: Acrelândia, Assis Brasil, Brasiléia, Bujari, Cruzeiro do Sul, Feijó, Rio Branco and Xapuri. In these, 116 species of 15 genera of sand flies were identified. Therefore, according to all the studies carried out on phlebotomine fauna, the state of Acre has 116 reported species.

The genera with most species were: *Evandromyia* (18), *Psychodopygus* (18), *Psathyromyia* (17), *Lutzomyia* (10), *Nyssomyia* (9), *Trichophoromyia* (9), *Micropygomyia* (9), *Pintomyia* (6), *Pressatia* (4), *Sciopemyia* (4), *Bichromomyia* (3), *Brumptomyia* (3), *Trichopygomyia* (2), *Viannamyia* (2), *Migonemyia* (1). Note that 12 observations were reported as *Trichophoromyia* sp., because females of the two species *Trichophoromyia auraensis* (Mangabeira, 1942) and *Trichophoromyia ruifreitasi* Oliveira, Teles, Medeiros, Camargo & Pessoa 2015 cannot be distinguished by morphology.

In terms of diversity, the municipalities with the most species recorded were: Assis Brasil (78), Rio Branco (68), Xapuri (58), Bujari/Xapuri/Rio Branco (50), Cruzeiro do Sul (30), Brasiléia (20), Feijó (14), Acrelândia (12) (table in GigaDB [1]).

Note that the number of species for Rio Branco and Xapuri may be different from that shown here (table in GigaDB [1]). This is because one of the articles clustered the results for both municipalities. Therefore, we had to group all species records (for this specific article) under the same name (Bujari/Rio Branco/Xapuri) (table in GigaDB [1]). The species *Psychodopygus corossoniensis* (Le Pont & Pajot, 1978) was only recorded once in the state, but because the author did not name the municipality in which specimens were collected (the only location reference is "AC Highway Km 22"), the county for this observation is NA.

The chronological table provided in GigaDB [1] presents the scientific data used to compile the dataset by municipality, author and species names.

## CONTEXT

Phlebotomine sand flies (Diptera, Psychodidae) are insects of great medical interest since they can transmit pathogens such as leishmaniases, bartonellosis and some arboviruses [2]. In the Americas, 547 sand fly species have been recorded, with most in tropical areas. The Amazon region shows the greatest diversity and species richness for these insects [3, 4]. Located in the Amazon region, the state of Acre exhibits great richness of phlebotomine sand fly species, with recent reports of new records and descriptions of new species [3–7].

The first published study was carried out by Martins and Silva [8] on the sand fly fauna in the capital Rio Branco. They recorded 30 species, with *Pintomyia nevesi* (Damasceno & Arouck, 1956) being dominant. In the early 1980s, Arias and Freitas [9] carried out research in the municipalities of Cruzeiro do Sul, Feijó and Rio Branco, finding 50 species, with *Trichophoromyia auraensis* the most frequent. In the late 2000s, Azevedo *et al.* [10] conducted research in the municipalities of Rio Branco, Bujari and Xapuri, and found a predominance of *Nyssomyia whitmani* (Antunes & Coutinho, 1939), *Nyssomyia antunesi* (Coutinho, 1939) and *Th. auraensis*. At the same time, in rural areas of the municipality of Acrelândia, Silva-Nunes *et al.* [11] found a predominance of *Ny. whitmani* and *Ny. antunesi*. Similar observations have also been made in peri-urban and forest areas of Rio Branco [7, 12]. In Assis Brasil, 67 species were collected, with three new records for Acre, *Evandromyia georgii* (Freitas & Barret 2002), *Lutzomyia evangelistai* (Martins & Fraiha 1971) and *Psychodopygus complexus* (Mangabeira, 1941), with the most abundant species being *Trichophoromyia* spp. (*Th. auraensis*/*Th. ruifreitasi*) and *Psychodopygus davisi* (Root, 1934) both found with *Leishmania braziliensis* (Vianna, 1911) and *Leishmania guyanensis* (Floch, 1954) by molecular techniques [13]. In this same locality, two species were described: *Lutzomyia naiffi* (Teles *et al.* 2013) and *Th. ruifreitasi* (Oliveira *et al.* 2015), with the latter females being indistinguishable from several others of the genus *Trichophoromyia*. In addition, Assis Brazil has recently unveiled new records of sand fly species [14–16].

Recent studies carried out on the Brazilian–Bolivian border (Brasiléia and Xapuri municipalities) highlighted the richness and diversity of sand fly species, with *Nyssomyia shawi* (Fraiha, Ward & Ready, 1981) and *Trichophoromyia* sp. being the most frequent. These species were also found to be infected with *Leishmania* DNA [4]. In addition, these studies verified the occurrence of the great diversity of species vectors, captured in both domiciliary and forest environments. Noteworthy in this locality is the description of a new species and the revalidation of some taxa [17–19].

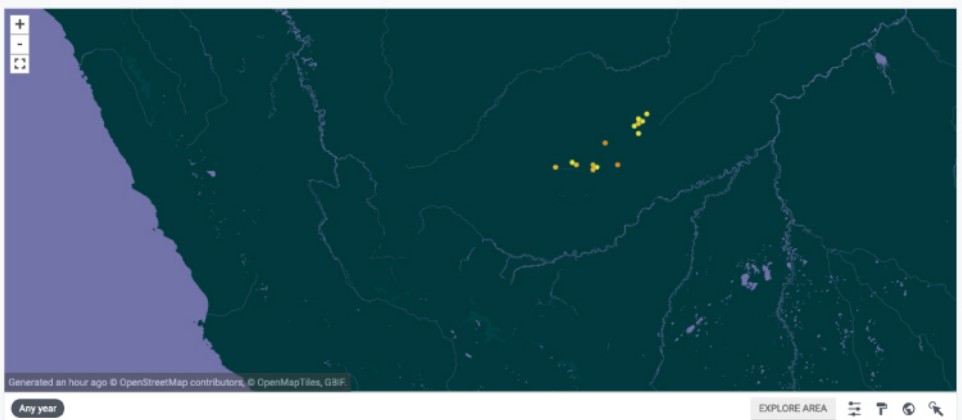

**Figure 1.** Map of the georeferenced occurrences hosted by GBIF [24]. https://www.gbif.org/dataset/f0ccf436-e2e8-4f61-af19-753cdb73ca00

Acre state is a hotspot of American tegumentary leishmaniasis in the Amazon biome. It affects people regardless of their gender or age; however, there is an increase in notifications in women and children, which suggests different transmission cycles occurring in the region [20, 21]. All species of the *Leishmania* parasite subgenus *Viannia* have been described as circulating in humans in this region (with the exception of *Leishmania lindenbergi* Silveira, Ishikawa & de Souza, 2002) and *Leishmania amazonensis* (of the subgenus *Leishmania*) [13, 21, 22].

## METHODS

### Study area

The state of Acre state is situated in the north of Brazil and is part of the Amazon region, corresponding to 1.92% of the Brazilian territory. The population is about 900,000 inhabitants. The economy is based mainly in the extraction of forest materials, particularly rubber and Brazilian chestnut for exportation. Its vegetation is tropical forest, and it has humid and hot equatorial climate. The average annual temperature is 31.5 °C and rainfall is 2100 mm [23].

Geographically, the state is divided into two meso-regions named Juruá and Acre Valley, which are subdivided into five microregions: Rio Branco, Sena Madureira and Brasiléia belonging to Acre Valley, and Cruzeiro do Sul and Tarauacá belonging to Juruá Valley [23] (see Figure 1).

### Preserved specimens

The records of two preserved sand fly specimens were included in the dataset. These insects were captured using CDC/Shannon traps. After capture, the insects were screened, separated, and identified along with their capture location. The insects were identified using the procedure proposed by Forattini [25]. After the identification process, specimens were mounted between slide and cover slip in Enece medium [26] and identified according to Galati [3].

**Table 1.** All variables used in the dataset with its name, details and how they were subset.

| Variable name | Variable detail | Variables subset |
|---|---|---|
| datasetName | Name of the dataset | Dataset name and information of the revised data |
| basisOfRecord | Type of material used to obtain the data | |
| bibliographicCitation | DOI or link for the citation used | |
| individualCount | Number of specimens recorded | Condition and quantity of captured sand flies |
| sex | Sex of the specimen | |
| lifeStage | Development stage of captured specimens | |
| preparation | Type of mount solution used to preserve the specimens | Capture: location, type of trap and preparation details |
| samplingProtocol | Type of trap used to capture the insects | |
| samplingEffort | Total time spent capturing in hours | |
| habitat | Type of environment where the trap was located | |
| continent | Continent where the study was conducted | |
| country | Country where the study was conducted | |
| countryCode | International code for the country | |
| stateProvince | State where the study was carried out | |
| county | Municipality where the study was carried out | |
| locality | Additional information about the location where the insect was captured | |
| locationRemarks | Specific area or condition where the insect was captured | |
| decimalLatitude | Capture site latitude in decimal degrees | |
| decimalLongitude | Capture side longitude in decimal degrees | |
| identifiedBy | Name of author(s) that identified the specimens | Species Identification: name of authors, year of identification and taxonomic detail of sand fly specimens |
| dateIdentified | Year when the record was published | |
| scientificName | Species name and authorship | |
| kingdom | Kingdom that the species belongs to | |
| phylum | Pphylum that the species belongs to | |
| class | Class that the species belongs to | |
| order | Order that the species belongs to | |
| family | Family that the species belongs to | |
| genus | Genus that the species belongs to | |
| subgenus | Subgenus that the species belongs to | |
| specificEpithet | Species-specific name | |
| infraspecificEpithet | Species sub-specific name | |
| taxonRank | Taxonomic level of the specimen | |
| ScientificNameAuthorship | Author and year of species description | |

## Bibliographic review

To review bibliographic material, the following online databases were used: PubMed, Google Scholar, SciELO, BVS - Biblioteca Virtual em Saúde. The following search terms were used: "Acre" AND ("sand fly" OR "sandfly" OR "sandflies" OR "sand flies" OR "Phlebotominae"). Bibliographic references cited in scientific articles were also used as data sources.

All scientific articles/books were assessed to obtain data from 33 standardized variables of Darwin Core terms [27] (Table 1). These variables were grouped into four subsets to identify the dataset; to describe essential information about specimen condition and quantity; capture location and methods; and taxonomic status for each observation (Table 1).

## DATA VALIDATION AND QUALITY CONTROL

Data were collected and checked with the aid of other bibliographical references [3]. Names were checked by experienced taxonomists, and data were validated via the GBIF data validator tool upon submission of the data [24].

## REUSE POTENTIAL

We have assembled the most exhaustive scientific data on sand flies in Acre, Brazil that have been published until now. This dataset provides important knowledge on the distribution, identification, and taxonomic status of the sand fly species already recorded in

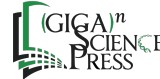

the state, and will form a solid reference for future studies on sand fly ecology, epidemiology and taxonomy in the area.

## DATA AVAILABILITY

Phlebotomine sand fly species registered by municipality in Acre state, and chronological records of phlebotomine sand fly species by author and municipality (built from the cited literature and [27–32]) are both available in the *GigaScience* GigaDB repository [1]. The dataset used in this manuscript was deposited in the Sistema de Informação sobre a Biodiversidade Brasileira (SiBBr) integrated publishing toolkit (IPT) [24].

## EDITOR'S NOTE

This paper is part of a series of Data Release articles working with GBIF and supported by the Special Programme for Research and Training in Tropical Diseases (TDR), hosted at the World Health Organization [33].

## LIST OF ABBREVIATIONS

IPT: Integrated publishing toolkit; SiBBr: Sistema de Informação sobre a Biodiversidade Brasileira; TDR: Special Programme for Research and Training in Tropical Diseases

## ETHICAL APPROVAL

Not applicable.

## CONSENT FOR PUBLICATION

Not applicable.

## COMPETING INTERESTS

The authors declare that they have no competing interests.

## FUNDING

Not applicable.

## AUTHORS' CONTRIBUTIONS

REG: compilation, organization of data, and writing of the manuscript; AJA: data revision and writing of the manuscript; PHFS: data revision and writing of the manuscript; AFB: sample collecting, identification, data revision and writing of the manuscript.

## ACKNOWLEDGEMENTS

We would like to thank Clara Baringo Fonseca for her help in the preparation of the Darwin Core spreadsheet.

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
