## [Reviewer Report]

Upload additional filesDRR-202202-09/form/DRR-202202-09_Data-Review-MAT.pdfReviewer name and names of any other individual's who aided in reviewer Mary Ann TuliDo you understand and agree to our policy of having open and named reviews, and having your review included with the published papers. (If no, please inform the editor that you cannot review this manuscript.)YesIs the language of sufficient quality?YesPlease add additional comments on language quality to clarify if needed
Are all data available and do they match the descriptions in the paper? NoAdditional CommentsSpecies column in the GBIF download: 814 (out of 1096) records are curated to species level. The numbers in the MS do not correspond to what is in GBIF e.g. MS states there are 4 Pressatia species, yet none in GBIF are curated to species level. There are 16 records curated as Genus Pressatia. Should I be looking in the Scientific Name column? It is not clear. 
Are the data and metadata consistent with relevant minimum information or reporting standards? See GigaDB checklists for examples <a href="http://gigadb.org/site/guide" target="_blank">http://gigadb.org/site/guide</a>YesAdditional CommentsIs the data acquisition clear, complete and methodologically sound?YesAdditional CommentsIs there sufficient detail in the methods and data-processing steps to allow reproduction?YesAdditional CommentsIs there sufficient data validation and statistical analyses of data quality? YesAdditional CommentsIs the validation suitable for this type of data?YesAdditional CommentsIs there sufficient information for others to reuse this dataset or integrate it with other data?YesAdditional CommentsAny Additional Overall Comments to the Authorexcel tables (1,2,3) need to be uploaded in machine readable format.RecommendationMinor Revision

---

## [Reviewer Report]

Upload additional filesDRR-202202-09/form/Sand flies Acre literature Review dataset ISP.docxReviewer name and names of any other individual's who aided in reviewer Israel de Souza PintoDo you understand and agree to our policy of having open and named reviews, and having your review included with the published papers. (If no, please inform the editor that you cannot review this manuscript.)YesIs the language of sufficient quality?YesPlease add additional comments on language quality to clarify if needed
Are all data available and do they match the descriptions in the paper? YesAdditional CommentsAre the data and metadata consistent with relevant minimum information or reporting standards? See GigaDB checklists for examples <a href="http://gigadb.org/site/guide" target="_blank">http://gigadb.org/site/guide</a>YesAdditional CommentsIs the data acquisition clear, complete and methodologically sound?YesAdditional CommentsIs there sufficient detail in the methods and data-processing steps to allow reproduction?YesAdditional CommentsIs there sufficient data validation and statistical analyses of data quality? YesAdditional CommentsIs the validation suitable for this type of data?YesAdditional CommentsIs there sufficient information for others to reuse this dataset or integrate it with other data?YesAdditional CommentsAny Additional Overall Comments to the AuthorThe manuscript DRR-202202-09 by Godoy et al. addresses valuable information regarding sand fly fauna from state of Acre, Brazil. The manuscript needs a minor revision before be published. See my suggestions in the manuscript text. RecommendationMinor Revision